# Surface Modification of Silicone by Dielectric Barrier Discharge Plasma

**DOI:** 10.3390/ma16082973

**Published:** 2023-04-08

**Authors:** Krzysztof Krawczyk, Agnieszka Jankowska, Michał Młotek, Bogdan Ulejczyk, Tomasz Kobiela, Krystyna Ławniczak-Jabłońska

**Affiliations:** 1Faculty of Chemistry, Warsaw University of Technology, ul. Noakowskiego 3, 00-664 Warsaw, Poland; krzysztof.krawczyk@pw.edu.pl (K.K.); a.jankowska@secura.com.pl (A.J.); bogdan.ulejczyk@pw.edu.pl (B.U.); tomasz.kobiela@pw.edu.pl (T.K.); 2Institute of Physics, Polish Academy of Sciences, Al. Lotnikow 32/46, 02-664 Warsaw, Poland; jablo@ifpan.edu.pl

**Keywords:** surface modification, silicone, plasma, wettability

## Abstract

The objective of the study was to modify the surface of the silicone rubber, using dielectric barrier discharge (DBD) to improve its hydrophilic properties. The influence of the exposure time, discharge power, and gas composition—in which the dielectric barrier discharge was generated—on the properties of the silicone surface layer were examined. After the modification, the wetting angles of the surface were measured. Then, the value of surface free energy (SFE) and changes in the polar components of the modified silicone over time were determined using the Owens–Wendt method. The surfaces and morphology of the selected samples before and after plasma modification were examined by Fourier-transform infrared spectroscopy with attenuated total reflectance (FTIR–ATR), atomic force microscopy AFM, and X-ray photoelectron spectroscopy (XPS). Based on the research, it can be concluded that the silicone surface can be modified using a dielectric barrier discharge. Surface modification, regardless of the chosen method, is not permanent. The AFM and XPS study show that the structure’s ratio of oxygen to carbon increases. However, after less than four weeks, it decreases and reaches the value of the unmodified silicone. It was found that the cause of the changes in the parameters of the modified silicone rubber is the disappearance of oxygen-containing groups on the surface and a decrease in the molar ratio of oxygen to carbon, causing the RMS surface roughness and the roughness factor to return to the initial values.

## 1. Introduction

Plastics are one of the most important materials necessary in modern industry and everyday life. Their low price and good properties, such as high resistance to chemicals and low density, resulted in a rapid increase in their production [1]. In 2021, world production of plastics amounted to 367 million tons [2]. Polymeric materials are used primarily in the packaging, automotive, construction, medicine, and electrical wiring industries. Polyolefins, polyethylene terephthalate (PET), polyvinyl chloride (PVC), silicone, and butadiene rubber are most often used for these applications. Silicone rubber is a bio-compatible material used in many medical applications, especially for producing disposable materials such as catheters and drains [1,3]. Drains made of silicone rubber may remain in the patient’s body for up to 30 days without causing inflammation.

While manufacturing complete products, polymer materials undergo many technological procedures in the industry. Such systems include printing, gluing, or metalizing. Unfortunately, plastics’ smooth and difficult-to-wet surface causes problems when carrying out these processes [4,5]. The material’s surface is often contaminated with adsorbed water, migrating components of the material such as plasticizers, lubricants, and anti-burning agents, which lower the adhesion between the product and the substances applied to the surface [4]. The primary purpose of surface modification of polymer materials is to increase the adhesion between the surface of the material and the substance deposited on the surface. Surface modification can change surface free energy, increase or decrease hydrophilicity or hydrophobicity, remove impurities, improve electrical conductivity, and reduce the coefficient of friction. It is also possible to introduce specific functional groups on the surface to change their chemical properties or the ability to stain [6].

Methods of surface modification can be divided into physical and chemical. Physical methods consist of mechanical, plasma, and flame methods, as well as processes using high-energy electrons, gamma rays, laser light, and ions of various elements [7,8]. These also include methods such as vapor deposition or cathodic sputtering, which deposit a metallic layer on plastic.

Chemical methods usually involve immersing or spraying plastics in highly oxidizing liquids [9,10]. Chemical processes are easy to carry out, but their major disadvantages are the high water consumption and corrosion of the equipment in which the modification is carried out. Chemical methods are expensive and are being replaced by physical ones [11,12,13].

Plasma can alter many surface properties of polymeric materials, including chemical, electrical, optical, biological, and mechanical properties [14,15]. The use of this method can cause the following: cleaning the surface of impurities, etching (increasing roughness), introducing new functional groups to the surface layer (mainly polar groups, e.g., OH), grafting precursors (e.g., unsaturated monomers) on the surface, crosslinking the material surface [12], or charge dissipation [16].

Surface modification using plasma discharge is currently the most commonly used method for modifying polydimethylsiloxane [17,18].

The plasma generated in oxygen has been used in the process of plastic surface modification both in industry and for research purposes [19,20,21]. The oxidation of the surface layer and the formation of the following groups: C-O, C=O, O-C=O, C-O-O was detected, which increases wettability and free surface energy. This effect increases with increasing sample exposure time and the discharge’s power for polyolefins [22].

The use of plasma generated in nitrogen or ammonia atmosphere also leads to an increase in wettability, surface free energy (SFE), and adhesive strength, which cause the following groups to form in the surface layer: C-N, NH_2_, C-NH-C, and C_2_N-C [23,24,25].

In the plasma generated in the air, mainly C-O groups are formed in the surface layer [26]. In the plasma generated in gases containing chlorine atoms, an increase in the wettability of plastics, in particular polyolefins, is also observed, making them hydrophilic. These gases can be, for example, tetrachloromethane (CCl_4_), trichloromethane (CHCl_3_), or trifluoro chloromethane (CF_3_Cl) [27,28,29]. For polypropylene, the interaction of the plasma generated in the tetrachloromethane atmosphere leads to more significant changes in the SFE and the contact angle than in the oxygen, nitrogen, and argon plasma generated under the same conditions [30].

Using plasma generated in fluorine-containing gases increases the surface’s contact angle and hydrophobicity. This is caused by the exchange of hydrogen atoms in macromolecules with fluorine atoms [31]. The flashover properties of rubber poly(dimethylsiloxane) (PDMS) can be improved by introducing fluoro-carbon groups on its surface. This process was conducted by Wang et al. in RF plasma [32].

The modification of polypropylene using corona discharges increases surface roughness [33]. Regarding styrene-butadiene copolymers and rubbers, higher adhesion was obtained between these materials and polyurethane adhesives due to the formed C-O, C=O, and C-O-O groups. No changes in the morphology of the surface layer were observed [34].

Plasma techniques are environmentally friendly, leave no waste, and are inexpensive and fast. They can be used for products with various geometries and materials (metals, ceramics, polymers, and composites). Plasma easily penetrates holes, crevices, and other hard-to-reach places, allowing you to obtain a modified and sterile surface in one step. Plasma modification changes only the product’s thin surface layers (from several to several dozen nanometers) without affecting its internal structures. This allows the preservation of the product’s features that are important during its long-term use [35,36,37,38].

As a result of the plasma functionalization of silicone rubber, adhesion can be enhanced. However, due to the reorientation of polar groups, condensation of OH groups, and diffusion of oligomers from the bulk of the material to the surface layer, adhesion decreases over time [39,40]. With aging, the hydrophilicity caused by the oxidation of the surface of the plastics decreases. This process, called hydrophobic recovery, is a diffusion process [41].

In the presented paper, the dielectric barrier discharge (DBD) method was used to modify the surface of the silicone rubber-made PDMS. PDMS is widely used for a variety of applications in the medical area (catheters, aesthetic implants, and other disposable medical devices). PDMS is inert to body fluids and a wide range of medical fluids; it is biocompatible, thermally stable, has good mechanical properties, and is easily manufactured. The objective of the study was to modify the surface of the silicone rubber to improve its hydrophilic properties. A more hydrophilic surface facilitates the movement of the cannula or drains through the organism and reduces the possibility of unwanted biofilm formation. The influences of the exposure time, discharge power, and gas composition on the properties of the silicone surface layer were studied. The value of surface free energy and changes in the polar components were determined using the Owens–Wendt method. Atomic force microscopy (AFM), Fourier-transform infrared spectroscopy with attenuated total reflectance (FTIR-ATR), and X-ray photoelectron spectroscopy (XPS) examined selected samples’ morphology and surface composition before and after plasma modification. A novelty of the study was the conduction of modifications in the gas of different compositions and the determination of the rate and type of surface changes occurring over time.

## 2. Experimental

Silicone rubber granulate ELASTOSIL^®^ LR 3003/50 by Wacker Chemie A.G. used for products’ medical applications was studied. The aim of the work was to examine the influence of process parameters of the barrier discharge on the surface properties of poly (dimethylsiloxane) rubber.

### 2.1. Dielectric Barrier Discharge Reactor

The barrier discharge was generated in the reactor (Figure 1) under slightly reduced pressure (approx. 0.67 atm) and supplied an alternating current with a frequency of 8 kHz and a voltage of approx. 8 kV.

The barrier discharge was generated in several gases: air, nitrogen, oxygen, mixtures of carbon dioxide with argon or air, and air with nitrogen maintaining the 1:1 by volume ratios. The total gas flow rate was 10 L/min, set with Bronkhorst mass flow controllers (Figure 2). The reduced pressure was obtained with a vacuum pump (VP). The reactor was connected to a grounded high-voltage transformer (HVT). There was a frequency converter (FC) between the connection to the mains and the transformer. The discharge power was measured with an oscilloscope Tektronix DPO3034 China with Tektronix TCP0030 China current probes and Tektronix P6015 Beaverton, OR, USA voltage probes. Figure 3 shows an example of the current-voltage characteristics of the obtained barrier discharge.

### 2.2. Contact Angle Measurements

Measurements of contact angles were made using the Delta^®^ optical BioLight 200 Mińsk Mazowiecki, Poland apparatus and the ToupView software. The volume of a tiny droplet for all contact angle measurements was 3–4 mL. Deionized water with 1–3 μS/cm conductivity and stabilized diiodomethane > 99% were used as measuring liquids. After modification, the tested materials will be in contact with polar fluids (water and plasma); therefore, the polar component of silicone was investigated.

### 2.3. FTIR-ATR Measurements

For selected samples, FTIR-ATR spectroscopic analyses were carried out using the Thermo Scientific Nicolet 6700 Madison, WI, USA spectrophotometer. For each measurement, 64 scans with 8 cm^−1^ resolution were made. The Omnic program was used to analyze the results.

### 2.4. AFM Measurements

Sample surfaces were analyzed using AFM Park Systems XE-120 (Suwon, Republic of Korea) operating in a contact mode. At least two tips (PPP-CONTSCR, Nanosensors) for each sample were used to ensure reproducibility. All scans were performed under ambient conditions. Several images at different scan sizes and at various places for each sample were taken to gain better knowledge of the variations of local structures. For all images, measurements were started with the same values as the scan parameters. The scan rate was 1 Hz, and the set point was 1.31 nN. However, in each case, final optimizations were performed. From AFM measurements, it is possible to calculate the parameters which characterize surface roughening [42]. The following surface parameters were considered: the root-mean-squared surface roughness (RMS) and the roughness factor (the surface-to-projected-area ratio). Both parameters are important because they measure different features. RMS informs about the dispersion of height value in the selected region. It is taken from the local profiles, while the roughness factor (surface ratio) is the global parameter: it characterizes the surface development compared to the selected area (plane). The height distribution function was computed as the normalized histogram of height. The Gwyddion software was used for AFM data analysis.

### 2.5. XPS Investigation

The photoelectron spectra were recorded by a Prevac (Rogow, Poland) set-up equipped with a high-intensity monochromatic X-ray Al Kα (1486.69 eV) source Scienta MX 650, Scienta R4000 hemispherical analyzer, and charge neutralization. The silicone samples were fixed using double-sided conductive carbon tape on a sample holder. Experiments were performed with the X-ray source operated with a power of 50 W to avoid changes introduced by X-rays, pressure below 10^−9^ mbar, and to neutralize electron energy of 5 eV and a 20 mA current. To be sure that the charge is properly neutralized in the recorded spectra, care was taken not to keep the constant binding energy of the main C 1 s peak, and measurements of this line were repeated several times to be sure that the shape of the line is not changing with time, which would be evidence that the charge has not cumulated on the sample surface. Therefore, the binding energy of the C 1 s main line was not the same in all samples. This line was identified as a 2C-Si-O bond of C, and the energy scale of photoelectrons was calibrated to 285.0 eV for this line. The spectra were measured with an analyzer pass energy of 200 eV and 0.2 eV. With such a set, the optimal energy resolution to intensity was achieved. The full width at half maximum (FWHM) of the 4f_7/2_ Au line measured at the same experimental condition was 0.6 eV. Spectra were analyzed using the commercial CASA XPS software package (Casa Software Ltd., version 2.3.17) [43] with a Shirley background and a GL (30) line shape (70% Gaussian, 30% Lorentzian).

### 2.6. Research Methodology

The silicone rubber for medical purposes was tested. Samples of 3 × 1 cm were the object of the modification.

The surface wetting angles of water and diiodomethane were measured for the pristine sample and all modified samples. Three drops (six angles) for each of the liquids were tested. From the measurements, the average was calculated. After that, the polar and dispersive components, the change in the polar component, and the surface free energy were calculated using the Owens–Wendt equation. Polar or dispersive components for water and diiodomethane are 21.8 and 51 mJ/m^2^, and 48.5 and 2.3 mJ/m^2^, respectively [44].
∆*p* = pm − pn(1)
where ∆*p* is the change in the polar component, pm is the polar component after modification, and pn is the polar component before modification.

The contact angles of the unmodified silicone rubber surface, along with the calculated dispersion, polar components, and free surface energy, are presented in Table 1.

## 3. Results and Discussion

### 3.1. Modification of Silicone Rubber in Barrier Discharge

The influence of the exposure time (15, 30, and 60 s), the power of the barrier discharge (10, 20, and 30 W), and the composition of the gas in which the discharge was generated on the properties of the silicone surface layer were investigated. Initial tests were conducted with air. Further tests with oxygen, nitrogen, carbon dioxide, and argon mixtures were carried out based on the results. Gas mixtures contain 50% vol. air and 50% carbon dioxide or nitrogen. The discharge power was 10 and 30 W. The polar components and free surface energy of all modified silicone rubber are collected in Table 2.

By modifying the silicone rubber in the air, a considerable increase in the polar component and surface free energy was obtained, from 5.3 for pristine (Table 1) to 44.3 mJ/m^2^, and 36 to 73.5 mJ/m^2^, respectively. These values were obtained using the highest discharge power (30 W) and a 15 s exposure time (Table 2).

For all tested samples, the exposure times increased the change in polar component ∆*p* (Figure 4), which was in the range of 26.4 to 39 mJ/m^2^. The increase in the discharge power caused an increase in the polar component for all exposure times (Table 2). In the barrier discharge generated in the air, it was observed that for a constant discharge power, ∆*p* decreased with the increase of the exposure time. For 10 W, the value of ∆*p* decreases from 33.3 to 26.4 mJ/m^2^ with a rise of exposure time from 15 to 60 s. This is probably due to the lowering of the roughness produced in the first seconds of machining. Adding carbon dioxide (50%) to the air for the discharge power of 10 W resulted in a very high value of the polar component and SFE, 47.9 and 72 mJ/m^2^, respectively (Table 2). As in the case of the modification of silicone rubber in the air, for the power of 10 W, a decrease in ∆*p* was observed with increasing exposure time. For the higher power value (30 W), no effect of the exposure time on the change of the polar component ∆*p* was observed. It was 39–40.6 mJ/m^2^ (Figure 4). The highest value of the polar component (47.9 mJ/m^2^) was achieved by conducting the process for 15 s in a discharge of 10 W. By increasing the exposure time, a decrease in the polar component was observed, which reached the value of 42.7 mJ/m^2^ for 60 s (Table 2).

A similar effect was observed in the air + nitrogen mixture (50 vol.%). For 10 W, high values of the polar component were obtained: 44.2–48.3 mJ/m^2^ and SFE 69.4–73.0 mJ/m^2^ (Table 2). The maximum values of ∆*p* for the power of 10 and 30 W were only 0.4 mJ/m^2^ higher than after modification in the mixture of air and CO_2_ (Figure 4).

In pure oxygen, for 10 W, high values of the polar component and SFE were also obtained. They were 40.9–48.2 and 70.4–74.5 mJ/m^2^, respectively (Table 2). The change in the polar component compared to the unmodified silicone rubber was 35.6–41.2 for 10 W and 36.7–42.9 for 30 W (Figure 4). It was found that with exposure times of 15 and 30 s, the value of the polar component increases with the increasing discharge power. For an exposure time of 60 s, the value of the polar component and ∆*p* slightly decreased (Table 2, Figure 4). For an exposure time of 60 s and a power of 10 and 30 W, a decrease in the polar component was observed compared to 30 s. It was 2.9 and 6.2 mJ/m^2^ for the power of 10 and 30 W, respectively.

During the modification of silicone rubber in nitrogen, for a discharge power of 10 W, an increase of the polar component and ∆*p* is observed with increasing exposure time (Table 2, Figure 4). The increase in the polar component rose from 34.5 to 39.0 mJ/m^2^. No significant change in the polar component was observed for a discharge power of 30 W in nitrogen. The ∆*p* difference was approx. 2.3 mJ/m^2^ for exposure times of 30 and 60 s. The highest value of the polar component of 47.0 mJ/m^2^ (Table 2) was obtained for an exposure time of 30 s and a discharge power of 30 W.

Based on the obtained results, it was found that the maximum difference in the change of polar component (∆*p*) for the studies conducted in oxygen and nitrogen and the power of 10 and 30 W was 2.2 and 1.2 mJ/m^2^, respectively.

Variable changes in the silicone rubber’s contact angle were also observed during modification in a mixture of carbon dioxide and argon (Table 2). The maximum value of the polar component was 45.0 mJ/m^2^ (Table 2). For the discharge powers of 10 and 30 W and the exposure time of 60 s, the ∆*p* were 37.5 and 39.7 mJ/m^2^, respectively. These values are comparable with those obtained during modifications in other gases. The change in polar component ∆*p* (Figure 5) was only slightly lower than for pure nitrogen (39 mJ/m^2^) and oxygen (38.9 mJ/m^2^) for 10 W and a 60 s exposure time. For the power of 30 W and an exposure time of 60 s, the increase in the polar component (39.7 mJ/m^2^) was higher than that obtained in oxygen (36.7 mJ/m^2^) and nitrogen (39.4 mJ/m^2^).

### 3.2. The Durability of Modification over Time

The contact angles of the selected samples were examined immediately after modification and after a specified time (Table 3). In the meantime, the samples were stored in a desiccator in argon. Regardless of the gas in which the barrier discharge occurred, the values of the polar components of silicone samples decrease with time. Thus, the modifications with the use of the barrier discharge are not permanent. For samples modified in an oxygen atmosphere, the value of the polar component decreased by approx. 60% in one week after the modification. During the next ten days, the value of the polar component decreased by another 34% in relation to the values achieved on the modification date. The values of the polar components of the samples modified in the gas mixture of carbon dioxide and argon showed an approx. 93% decrease for two weeks from the modification. The exposure time does not affect the durability of the modification. For the sample modified for 10 s, there was an 83% decrease in the value of the polar component. For the 60-s modification, the decrease was 86%.

In summary, modification is most heavily reversed in the first few days, after which the rate of change slows down, finally reaching polar values about 90% lower than those achieved on the modification day. The durability of the modification is not dependent on the discharge power and the gas in which it was generated, nor the exposure time.

On this basis, it was found that the increase in silicone rubber’s polar components and SFE is due solely to the plasma’s and discharge power’s interaction. The results of FTIR confirmed this analysis of silicone samples modified in carbon dioxide and argon for 60 s and at a power of 10 W (Figure 5). On the spectra, peaks may be identified as follows: stretching and bending of C-H bonds at 2960 cm^−1^, Si-CH_3_ at 1250 cm^−1^, Si-O-Si bonds stretching at 1050 cm^−1^, and Si-C at 780 cm^−1^. The spectra correspond with that obtained by Guan et al. [16]. After modification, no new peaks appear on the spectra of the samples modified in the barrier discharge, proving that this method’s sensitivity is too low. The changes on the surface of the modified silicone rubber were not observed. The measured contact angles and the calculated values of the polar components show that the wettability of the tested materials increases after modification.

As a surface sensitivity spectroscopy, XPS is a perfect tool to study the changes in elemental composition and chemistry after modification and the aging process. In Table 4, the element content for the pristine sample and samples with different measurement times after a 60 s modification in Ar with CO_2_ and Ar are collected. The ratio of elements is also indicated to analyze the changes in element content. Considering that the amount of Si should not change during the modification, one can notice that the O/Si ratio increases considerably during the modification in Ar + CO_2_ from 0.71 to 1.28 and, after four weeks, slightly decreases to 1.18. Therefore, the content of oxygen increases during this modification. In the case of carbon, the ratio decreased from 1.46 to 1.13 and then expanded to 1.60 after four weeks was observed.

Moreover, the increase of ratio O/C after modification is evidence that oxygen is more reactive in the applied process. More information one can gain from surface chemistry analysis is presented in Table 4 and Figure 6 and Figure 7. The study was performed similarly to in [45]. In relation to the silicide chemical formula presented in Figure 6F, in the pristine rubber, two carbon atoms are bonded to Si, which is also connected to oxygen (marked 2C-Si-O) in each of the monomers. At the end of the polymer, three carbon are bonded to Si (3C-Si-O) (Figure 6F). Some carbons related to contamination may be connected with oxygen only (C-O or C=O). Only 2% of C in the pristine rubber is in the C-O bond. It radically changes after modification in Ar + CO_2_, where 24% of C forms a C-O bond and 1% of C forms a double bond with oxygen (C=O), thus confirming the creation of polar components. After four weeks, the amount of 3C-Si-O bonds increased to 40% from 16% in pristine, and C-O decreased to 6%, similar to what was observed in polar component measures (Table 5). The increased 3C-Si-O component is evidence that the polymer becomes much shorter after modification. Analyzing oxygen, we found that the dominant part is bonded to silicon (O-Si), and a single percentage with contamination. At the Si site, most of the atoms are in polymer (C-Si-O) bonds. After modification in Ar + CO_2,_ about 20% of Si forms the bond with energy characteristics for the Si-OH or SiO_x_ bond (Figure 7). This confirms they break off the polymer chain, as evidenced by the increase in the number of 3C-Si-O bonds.

Much fewer changes in rubber bonds chemistry were observed in modifying in Ar only. Small increases in the limit of errors (from 1.19 to 1.22) after four weeks were observed in the O/Si ratio, and a slight decrease (from 2.04 to 1.93) in the C/Si ratio. However, in both cases, it is a considerable increase compared to pristine rubber. Therefore, the rubber was modified, but after five days, it was stable. Moreover, we did not observe a significant rise in the number of 3C-Si-O bonds, so the polymer molecule was not destroyed much, and not as much as when Si-O/OH bonds were formed. In summary, XPS studies confirm the formation of C-O, C=O, and Si-O/OH groups after modification. These groups were more stable in the case of Ar plasma modification.

The increase in the value of surface free energy and the polar component after modification in the plasma results from increased content of the polar oxygen groups C-O (~286.8 eV) and C=O (~289 eV). Taking into account that the XPS measurements were carried out under high vacuum conditions, the gases physically adsorbed on the surface would be removed during the preparation of the samples for measurement.

Silicone rubber samples, modified in carbon dioxide and argon for 60 s using a discharge power of 30 W, were also evaluated using AFM. Figure 8 presents AFM images from randomly selected 20 × 20 μm^2^ areas of investigated samples. For each surface, several local profiles of various places on the sample were taken. These surface profiles demonstrate that the plasma treatment caused surface roughening to vanish in time.

The surface roughness expressed as root mean square (RMS), calculated as an average of the profile height deviations from the mean line of the untreated sample, is equal to 34.8 nm ± 2.6 nm (mean ± standard deviation), and the roughness factor is equal to 1.05. After plasma treatment, an increase in the RMS surface roughness was observed to be 65.4 nm ± 3.2 nm. The same trend was observed for the roughness factor values (1.18). After four weeks, the studied parameters almost returned to their initial values (the RMS decreased to 44.6 nm ± 2.6 nm and the roughness factor to 1.08).

From the histogram before plasma treatment (Figure 9A), we notice many heights that fall in the 40 to 160 nm tall category, but very few larger than 170 nm. The distribution is unimodal and right-skewed.

After plasma treatment, many heights fall in the 200 to 400 nm tall category, but very few are larger than 400 nm and smaller than 100 nm (Figure 9B). After four weeks, many heights fall in the 140 to 260 nm tall category, but very few are larger than 260 nm (Figure 9C). In both cases, the distribution is unimodal and left skewed.

## 4. Conclusions

Nonequilibrium plasma of the dielectric-barrier discharge is a convenient method for surface modification of silicon rubber. Carrier gas composition significantly influences the modification process. In the DBD reactor, surface free energy and the polar component were increased; however, this modification was not permanent. The results of the XPS and AFM measurements confirm the change in the surface composition and morphology due to the plasma treatment of silicone rubber. The XPS studies confirm the formation of C-O, C=O, and Si-O/OH bonds after the plasma treatment of silicone rubber. In the case of modification in Ar and CO_2,_ the changes in the chemical bonds of C, O, and Si were more pronounced than in the case of modification in Ar only.

Moreover, after five days and after four weeks, the chemistry of rubber modified in Ar does not change much, opposite to the modification in mixed gases. In the last case, the number of 2C-Si-O bonds decreases from 82% in pristine rubber to 56% and 54% after one day and four weeks, respectively. Moreover, 18% and 29.5 % of Si formed SiOx or Si-OH bonds outside of C-Si-O polymer bond, evidence of the destruction of rubber, also confirmed by increased 3C-Si-O bonds from 16% when pristine to 40% after 4 weeks from Ar + CO_2_ modification. The FTIR-ATR method is not sensitive enough to detect a change of concentration-functional groups on the surface of materials modified with plasma methods.

## Figures and Tables

**Figure 1 materials-16-02973-f001:**
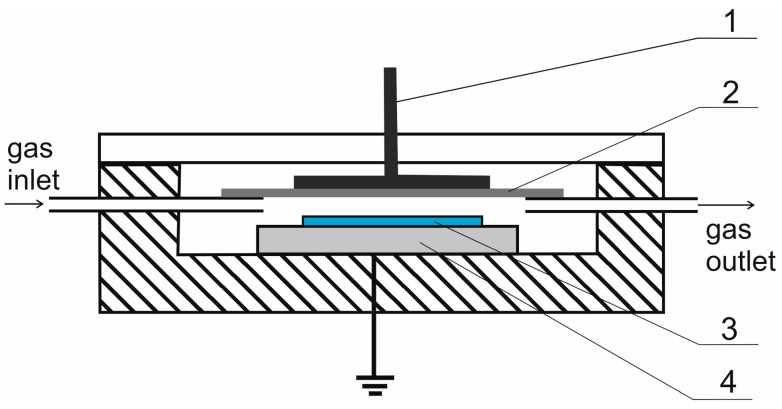
Diagram of a barrier discharge plasma reactor: 1—high voltage electrode, 2—quartz barrier, 3—rubber sample, 4—grounded electrode.

**Figure 2 materials-16-02973-f002:**
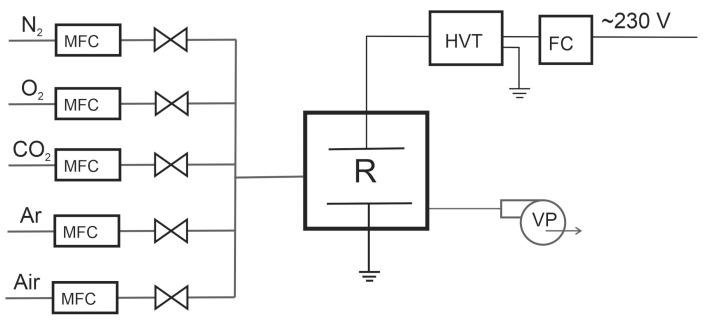
Apparatus diagram. MFC—flow controller, VP—vacuum pump, HVT—high voltage transformer, FC—frequency converter, R—reactor.

**Figure 3 materials-16-02973-f003:**
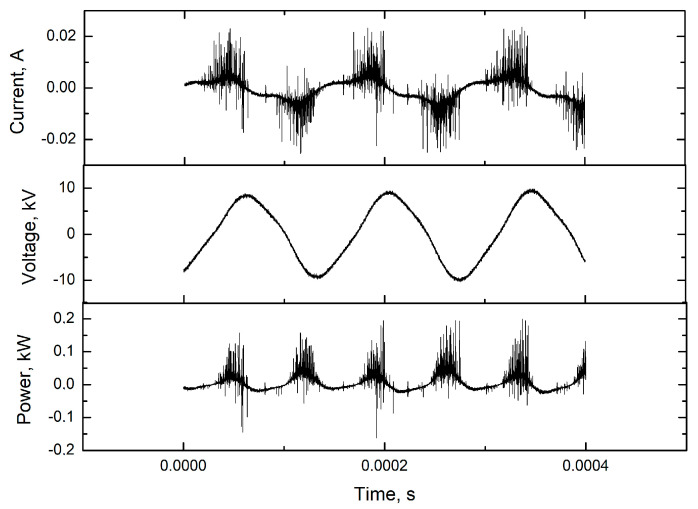
An example of a discharge current-voltage characteristic.

**Figure 4 materials-16-02973-f004:**
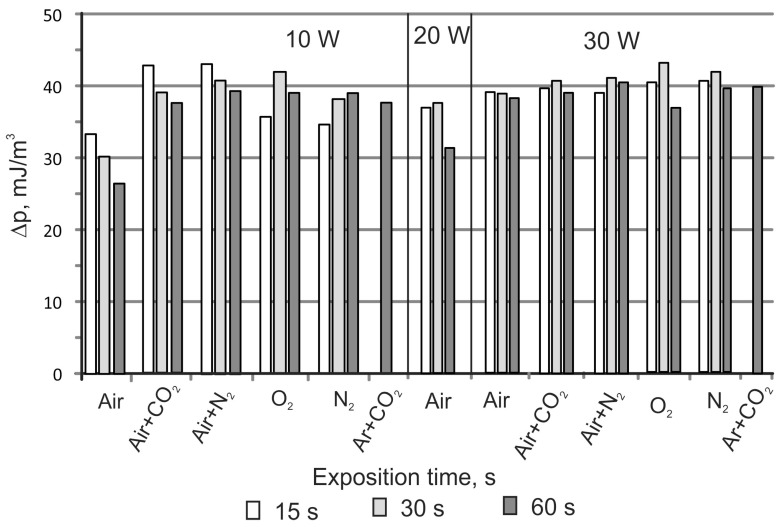
Effect of the gas composition, exposure time, and discharge power on the change in the polar component ∆*p*.

**Figure 5 materials-16-02973-f005:**
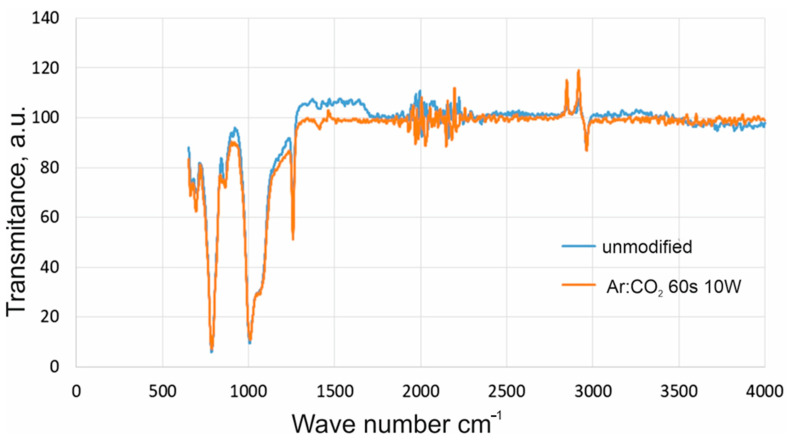
FTIR spectra of fresh and modified bonds in DBD silicone samples.

**Figure 6 materials-16-02973-f006:**
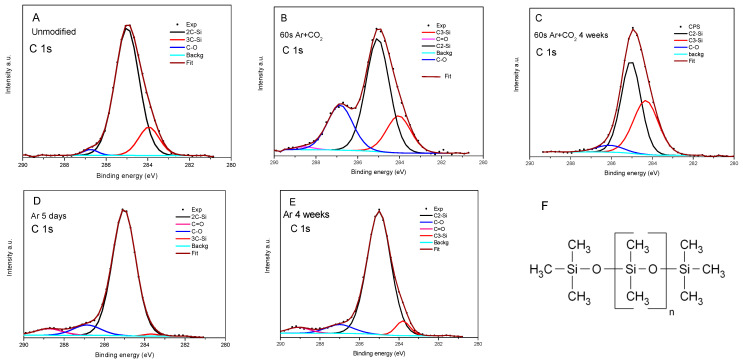
XPS 1s C spectra: (**A**)—unmodified silicone, (**B**)—directly after modification in Ar + CO_2_ for the 60 s, (**C**)—after four weeks, (**D**)—5 days after modification in Ar, (**E**)—after four weeks, (**F**)—chemical formula of silicone, where n is the number of repeating monomer.

**Figure 7 materials-16-02973-f007:**
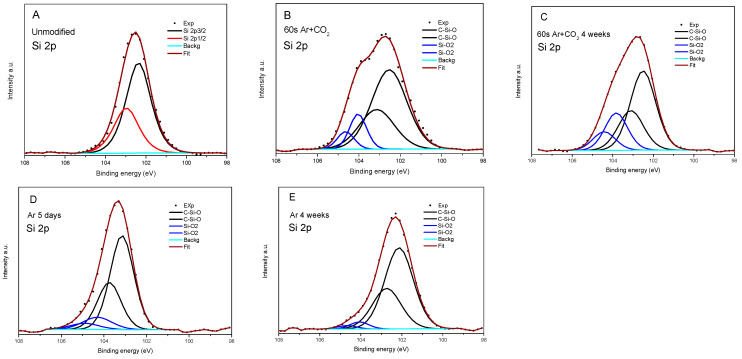
XPS 2p Si spectra: (**A**)—unmodified silicone, (**B**)—directly after modification in Ar + CO_2_ for the 60 s, (**C**)—after four weeks, (**D**)—5 days after modification in Ar, (**E**)—after four weeks.

**Figure 8 materials-16-02973-f008:**
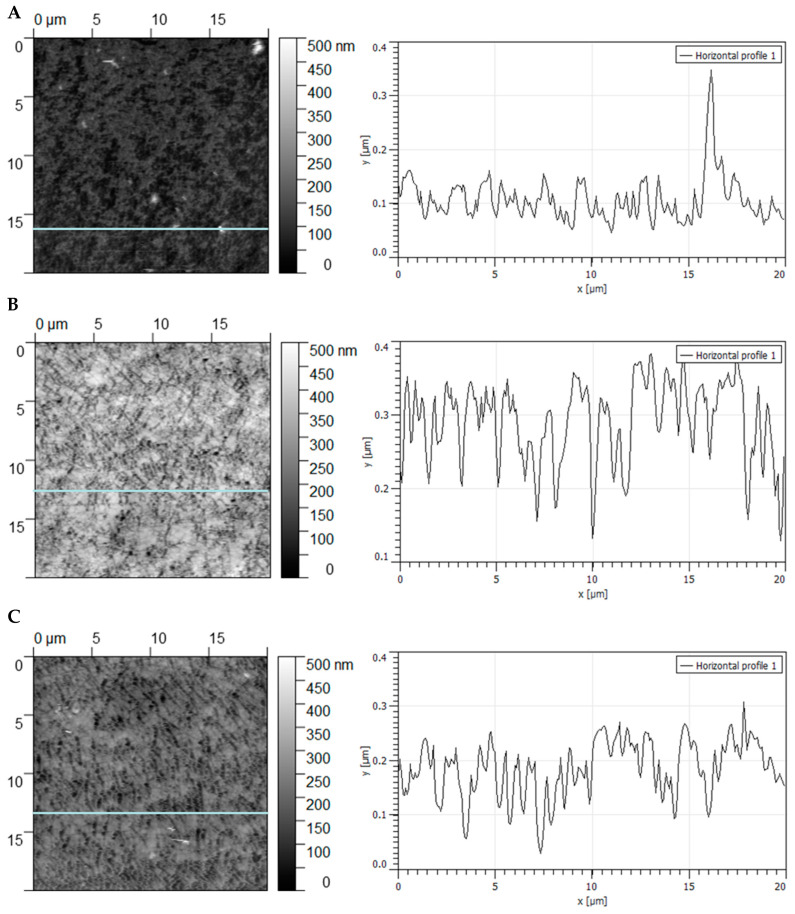
AFM images and surface profiles of silicon samples obtained: (**A**)—before plasma treatment, (**B**)—after plasma treatment, (**C**)—the same sample as in B stored for four weeks under ambient conditions.

**Figure 9 materials-16-02973-f009:**
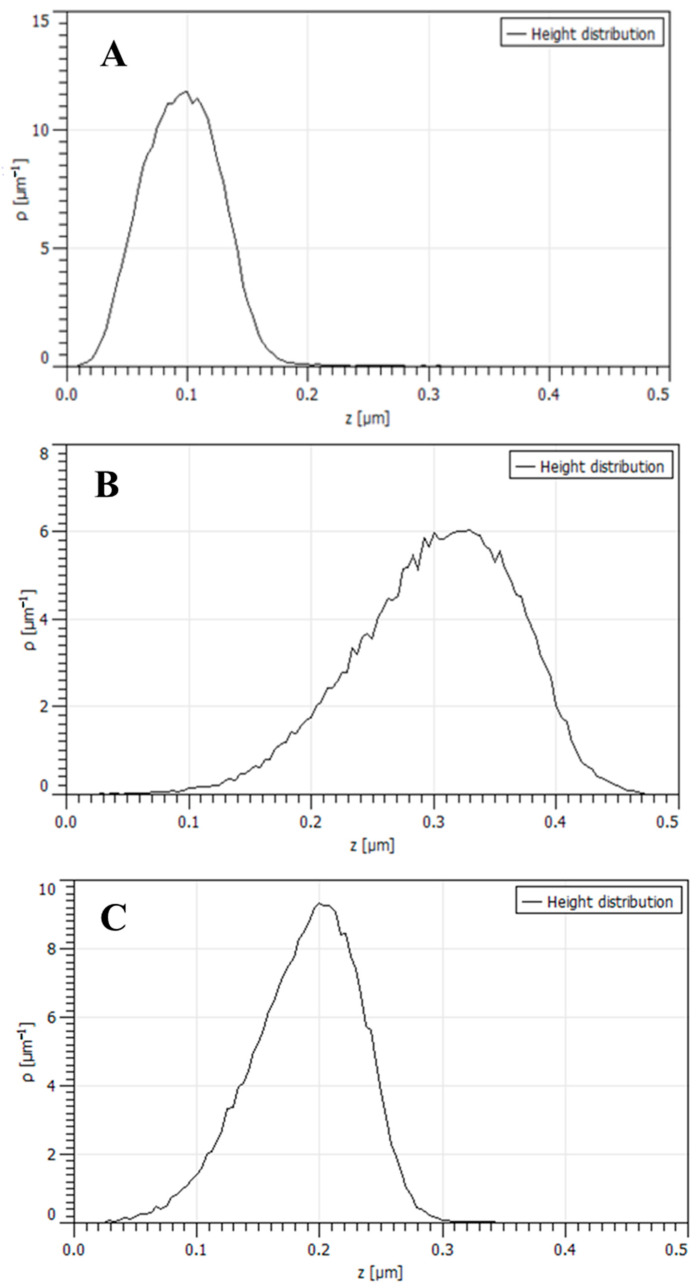
Histograms of the height distribution measured by AFM: (**A**)—before plasma treatment, (**B**)—after plasma treatment, (**C**)—the same sample as in B stored for four weeks under ambient conditions.

**Table 1 materials-16-02973-t001:** Contact angles, polar [pn] and dispersive components, and SFE of unmodified silicone rubber.

Unmodified Samples					
Material	Contact Angle Water [°]	Contact Angle Dijodomethan [°]	Dispersive Component [mJ/m^2^]	Polar Component (pn) [mJ/m^2^]	SFE [mJ/m^2^]
Silicone	80.7	48.9	30.8	5.3	36.0

**Table 2 materials-16-02973-t002:** Polar components [pm] and free surface energy SFE of the modified silicone rubber.

Exposition Time [s]	Power [W]	Air		Air + CO_2_		Air + N_2_		O_2_		N_2_		Ar + CO_2_	
		$[\mathrm{pm}]\;\left[\frac{mJ}{m^{2}}\right]$	SFE $\left[\frac{mJ}{m^{2}}\right]$	$[\mathrm{pm}]\;\left[\frac{mJ}{m^{2}}\right]$	SFE $\left[\frac{mJ}{m^{2}}\right]$	$[\mathrm{pm}]\;\left[\frac{mJ}{m^{2}}\right]$	SFE $\left[\frac{mJ}{m^{2}}\right]$	$[\mathrm{pm}]\;\left[\frac{mJ}{m^{2}}\right]$	SFE $\left[\frac{mJ}{m^{2}}\right]$	$[\mathrm{pm}]\;\left[\frac{mJ}{m^{2}}\right]$	SFE $\left[\frac{mJ}{m^{2}}\right]$	$[\mathrm{pm}]\;\left[\frac{mJ}{m^{2}}\right]$	SFE $\left[\frac{mJ}{m^{2}}\right]$
15	10	38.5	73.0	47.9	69.8	48.3	72.6	40.9	74.5	39.8	66.6	35.5	59.3
15	20	42.2	70.1	-	-			-	-	-	-	-	-
15	30	44.3	73.5	44.7	71.3	44.2	73.0	45.3	73.2	45.7	72.8	39.6	69.2
30	10	35.5	60.8	44.3	69.6	46.0	72.4	46.5	70.9	43.4	71.2	31.0	60.4
30	20	42.8	67.3	-	-			-	-	-	-	-	-
30	30	44.0	70.3	45.8	72.0	46.3	72.3	48.2	72.8	47.0	72.8	45.0	72.9
60	10	31.7	58.6	42.7	65.8	44.7	69.4	44.2	73.6	44.3	71.9	42.8	70.5
60	20	36.5	64.8	-	-			-	-	-	-	-	-
60	30	43.4	69.8	44.3	70.3	45.6	71.3	41.9	70.4	44.7	73.3	45.0	72.1

**Table 3 materials-16-02973-t003:** Contact angles and calculated polar, dispersion, and free surface energy components of samples modified with barrier discharge with the time at which re-measurements were taken.

Exposure Time [s]	Discharge Power [W]	Atmosphere	Time Elapsed Since Modification [weeks]	Polar Component $\left[\frac{mJ}{m^{2}}\right]$	$\mathrm{SFE}\;\left[\frac{mJ}{m^{2}}\right]$	Value Change SFE [%]	Changing the Value of the Polar Component [%]
15	30	pow.	0	44.7	73.5		
15	30	pow.	4	6.8	35.6	51	85
15	30	O_2_	0	45.3	73.2		
15	30	O_2_	1	18.2	47.1	36	60
15	30	O_2_	2	2.9	38.5	47	94
60	30	Ar.	0	41.1	70.2		
60	30	Ar	4	6.3	39.5	44	85
60	30	Ar + CO_2_	0	44.3	70.3		
60	30	Ar + CO_2_	4	3.1	42.6	39	93

**Table 4 materials-16-02973-t004:** The composition of the silicone rubber surface determined by the XPS [at%] and the ratio of elements.

Element	Pristine Rubber	Ar + CO_2_ 0 Day	Ar + CO_2_ Four Weeks	Ar 5 Days	Ar 4 Weeks
C	46.4	33.1	42.5	48.2	46.4
O	22.5	37.6	31.1	28.2	29.5
Si	31.8	29.3	26.4	23.6	24.1
O/C	0.48	1.14	0.73	0.59	0.64
O/Si	0.71	1.28	1.18	1.19	1.22
C/Si	1.46	1.13	1.60	2.04	1.93

**Table 5 materials-16-02973-t005:** Analysis of elements’ chemical environments and binding energy.

Sample	2C-Si-O	3C-Si-O	C-O	C=O	O-Si	O-C	C-Si-O	Si/OH/O
Pristine (BE eV) [%]	285.0 eV 82%	284.0; 16%	286.8 2%	-	532.6 98.5%	535.3 1.5%	102.4 100%	
Ar + CO_2_ 1 day	285.0 56%	284.0 19%	286.8 24%	288.6 1%	532.7 99%	535.5 1%	102.5 82%	104.0 18%
Ar + CO_2_ 4 weeks	285.0 54%	284.3 40%	286.1 6%	-	532.5 100%	-	102.5 70.5%	103.8 29.5
Ar 5 days	285.0 87%	283.6 1%	286.7 8%	288.6 4%	533.0 97%	534.5 3%	103.0 86%	104.2 14%
Ar 4 week	285.0 84%	283.8 6%	287.0 6%	289.0 4%	532.4 99%	535.6 1%	102.1 95%	104.1 5%

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
