# Peer review of "Surface Modification of Silicone by Dielectric Barrier Discharge Plasma"

_materials, 2023, doi:10.3390/ma16082973_

Round 1

Reviewer 1 Report

This research has the potential for publishing after improving the structural and scientific comments.

The following are some specific comments and points for consideration:

1.       Insert the full name of “PET, PVC” in the introduction part. Also, insert the full name of "FTIR – ATR, AFM, and XPS" in abstract part.

2.      Add relevant reference (s) for “Unfortunately, plastics' smooth and difficult-to-wet surface causes problems when carrying out these processes.”

3.      Insert the full name of AFM, FTIR-ATR, and XPS” in the introduction section.

4.      The type/quality of the presented data in Figure 3 should be improved. Use Excel, Origin lab, or Mathlab for the presentation of the discharge current-voltage characteristic curves.

5.     How much was the size of a tiny droplet for all contact angle measurements? Is it pure water?

6.      Insert the Z scale bar for AFM images in Figure 9.

7.      I highly recommend to the authors use the histogram distribution of surface roughness for all AFM maps to better distinguish the impact of the plasma treatment on the silicon surface before and after application.

8.      The authors should mention the massage of this research in a correct way especially in medicine alongside some suggestions related to the silicon surface treatment. Add this information in the conclusion and results/discussion parts.

Author Response

Dear Reviewer,

         First of all, I would like to thank You for Your comments and suggestions concerning my article.

Please find enclosed a paper entitled ”Surface modification of silicone by dielectric barrier discharge plasma” by Krzysztof Krawczyk, Agnieszka Jankowska, Michał Młotek, Bogdan Ulejczyk, Tomasz Kobiela, Krystyna Ławniczak-Jabłońska. All the reviewer’s suggestions have been considered when preparing this version of the manuscript. It has been attempted to improve the clarity of the article notably. I hope that the corrected manuscript is suitable for publication in the Materials.

Reviewer 2 Report

See the attached document.

Author Response

(The authors gave the same response as above.)

Reviewer 3 Report

This manuscript studied the influence of the plasma gas ambient and the power on the surface properties of silicon rubber. The modification was studied by FTIR – ATR, AFM, XPS, and Owens-Wendt method in various aspects. However, to improve the current manuscript, I believe the following comments should be addressed:

Comments 1): What is the variety of the Contact angles, polar [pn] and dispersive components, and SFE of unmodified silicone rubber? The trends of Polar components [pm] and free surface energy SFE of the modified silicone rubber in Table 2 are different for different gas ambient. Would the variation of the silicone rubber contribute to the scattered trend?

Comment 2): Considering that silicone rubber is not conductive, a charge neutralizer is used during XPS scans. Therefore, care must be taken while analyzing the chemical states in O 1s, Si 2p, and C 1s. Please elaborate on the methods of spectrometer calibration and the identification of the chemical states. For example, in Figure 7E, the C-Si-O is at ~ 100 eV, while in Figure 7D, it is at 101 eV. The peak positions of the two states are too far away to be assigned as the same state.

Comments 3): Please address the issue below:

1.      Line 15, DBD’s full name is duplicated.

2.      Line 72, the full name of SFE is missing.

3.      Figure 7c is missing? 

Author Response

(The authors gave the same response as above.)

Round 2

Reviewer 1 Report

No comments

Reviewer 2 Report

Dear authors,

Thank you very much for your responses.

Once again please do not explain the abrreviations twice.

Reviewer 3 Report

The authors have revised the manuscript accordingly. I suggest accepting the current version.